# Simultaneous Determination of Pharmaceuticals by Solid-phase Extraction and Liquid Chromatography-Tandem Mass Spectrometry: A Case Study from Sharjah Sewage Treatment Plant

**DOI:** 10.3390/molecules24030633

**Published:** 2019-02-11

**Authors:** Mohammad H. Semreen, Abdallah Shanableh, Lucy Semerjian, Hasan Alniss, Mouath Mousa, Xuelian Bai, Kumud Acharya

**Affiliations:** 1College of Pharmacy, University of Sharjah, P.O. Box 27272, Sharjah, United Arab Emirates; halniss@sharjah.ac.ae; 2Sharjah Institute for Medical Research, University of Sharjah, P.O. Box 27272, Sharjah, United Arab Emirates; 3Research Institute of Sciences and Engineering, University of Sharjah, P.O. Box 27272, Sharjah, United Arab Emirates; shanableh@sharjah.ac.ae (A.S.); mmousa2@sharjah.ac.ae (M.M.); 4College of Health Sciences, University of Sharjah, P.O. Box 27272, Sharjah, United Arab Emirates; lsemerjian@sharjah.ac.ae; 5Desert Research Institute, 755 E Flamingo Rd, Las Vegas, NV 89119, United States; xuelian.bai@dri.edu (X.B.); kumud.acharya@dri.edu (K.A.)

**Keywords:** liquid chromatography-tandem mass spectrometry, pharmaceuticals, wastewater analysis, solid phase extraction

## Abstract

The present work describes the optimization and validation of a highly selective and sensitive analytical method using solid phase extraction and liquid chromatography tandem mass spectrometry (SPE LC-MS/MS) for the determination of some frequently prescribed pharmaceuticals in urban wastewater received and treated by Sharjah sewage treatment plant (STP). The extraction efficiency of different SPE cartridges was tested and the simultaneous extraction of pharmaceuticals was successfully accomplished using hydrophilic-lipophilic-balanced reversed phase Waters^®^ Oasis HLB cartridge (200 mg/ 6 mL) at pH 3. The analytes were separated on an Aquity BEH C18 column (1.7 µm, 2.1 mm × 150 mm) using gradient elution and mass spectrometric analysis were performed in multiple reactions monitoring (MRM) selecting two precursor ions to produce ion transition for each pharmaceutical using positive electrospray ionization (+ESI) mode. The correlation coefficient values in the linear calibration plot for each target compound exceeded 0.99 and the recovery percentages of the investigated pharmaceuticals were more than 84%. Limit of detection (LOD) varied between 0.1–1.5 ng/L and limit of quantification (LOQ) was 0.3–5 ng/L for all analytes. The precision of the method was calculated as the relative standard deviation (RSD%) of replicate measurements and was found to be in the ranges of 2.2% to 7.7% and 2.2% to 8.6% for inter and intra-day analysis, respectively. All of the obtained validation parameters satisfied the requirements and guidelines of analytical method validation.

## 1. Introduction

As population growth rate escalates, greater demand is being placed on securing adequate water supply, and greater challenges arise in purifying wastewaters for reuse. The accumulation of commonly used pharmaceuticals such as antibiotics, analgesics, and antidepressants in water resources highlights the importance of investigating this type of contaminant [1,2,3,4,5,6,7]. The presence of these contaminants in waters, especially in drinking water, has become a major subject of worldwide growing concern, as such contaminants endanger human health safety and quality of water recourses [8,9,10].

One approach to augment available water supplies is through the use of treated wastewater for irrigation. However, local wastewater reuse regulations focus on conventional contaminants, such as biochemical oxygen demand (BOD_5_), chemical oxygen demand (COD), total suspended solids (TSS), nitrogen (N), phosphorus (P), total dissolved solids (TDS), and pathogens, but neglect emerging contaminants of concern (ECC) [11]. This necessitates the need to determine the behavior and regulate the concentrations of pharmaceutical contaminants during water recycling and reuse to safeguard public health and the environment as well as to reduce impediments to public acceptance of such alternative water management strategies. The possible presence of ECC, such as pharmaceuticals, illicit drugs, human hormones and personal care products [12], in reclaimed municipal wastewater is a growing challenge faced by developed and developing countries. Emerging contaminants raised concerns in recent years because of their potential chronic toxicity and development of bacterial pathogen resistance in humans and ecosystems [13,14]. Significant concentrations of ECC have been detected in sewage effluents worldwide, as many of these compounds may pass through conventional treatment systems without removal [15,16]. Treated wastewater reuse is a commonly practiced water management strategy in United Arab Emirates (UAE) to alleviate the country’s water shortage standing. The assessment of environmental and human risk of ECC in UAE is not well-established. Low concentration of pharmaceuticals (ng/L-µg/L) are considered as environmental contaminants, and their incomplete removal from wastewater by the treatment system represents a trigger for environmental and health risks [17,18]. Thus, there is a local uncertainty over the risk of human exposure and environmental contamination from ECC in wastewater reuse as well as the efficiency of existing wastewater treatment technologies for ECC removal.

The simultaneous determination of pharmaceuticals in wastewater is a challenging task due to the variation in their physicochemical properties, their low concentrations, and complexity of the environmental matrices [19,20,21,22,23,24,25]. Based on the literature surveys regarding the most commonly prescribed medications in the Emirate of Sharjah, it was found that the percentage of prescribing antibiotics to treat different types of infections was 45% [26]. Moreover, other studies showed high consumption of analgesic and antipyretic drugs (e.g., paracetamol) and beta blockers (e.g., Metoprolol) for the treatment of chronic diseases such as hypertension and heart failure [27]. A range of commonly prescribed pharmaceuticals including antibiotics, β-Blocker, Sulfa drugs, analgesics and antipsychotic were therefore selected to be investigated in this study; their pharmacological uses, chemical structures and physicochemical properties are summarized in Table 1.

Different analytical techniques have been used to investigate the presence of pharmaceuticals in wastewater, indicating levels of contamination in the range of ng/L-µg/L [28,29]. Previous studies employed solid phase extraction [30] and liquid-liquid extraction [31] prior to separation of the target compounds to minimize matrix interferences that might affect the detection and quantification. Some reported detection methods for pharmaceuticals include UV [32,33,34] and fluorescence detection [35]. However, the majority of previous related studies have employed liquid chromatography coupled with tandem mass spectrometry (LC-MS/MS) [36,37,38,39,40,41,42,43] and/or gas chromatography with tandem mass spectrometry [44,45], which are classified as the most powerful analytical techniques in term of sensitivity and selectivity compared with other commonly employed analytical techniques, allowing the detection and quantification of contaminants in wastewaters at levels of ng/L.

As the environmental and human health risks of ECC in UAE are not well established, the proposed research aims to develop a validated, highly sensitive and selective UPLC-MS/MS method based on solid phase extraction (SPE) to investigate the occurrence and amounts of antibiotics and other pharmaceuticals in wastewater plant in the city of Sharjah.

To our knowledge, this is the first study conducted in the city of Sharjah using an in-house developed LC-MS/MS method, which has proved to be highly sensitive with a limit of detection of 0.1–1.5 ng/L and a limit of quantification (LOQ) in the range of 0.3–5 ng/L for all analytes under investigation. Such a sensitive method is highly helpful in the assessment and continuous monitoring of the emerging contaminants of concern in the Sharjah sewage treatment plant. Findings will be of significance towards supporting decisions to optimize wastewater treatment and reuse strategies, as well as safeguard public health and the environment.

The proposed methodology presented in this study can be used to investigate complex wastewater types encountered in the City of Sharjah as the central wastewater treatment plant receives mainly domestic but also commercial and industrial wastewaters. It is of utmost importance to establish such a reliable methodology that can be adopted in local settings as to date, no studies were reported on the occurrence of ECCs in wastewaters generated in Sharjah, although the practice of wastewater reuse is highly encouraged and implemented in the United Arab Emirates as an initiative towards sustainable use of resources as well as alleviation of the country’s water shortage standing.

Development of analytical capabilities in the local settings will enable accurate measurements of ECCs in the UAE environment; the obtained results will provide vital information for analyzing the local risks posed by such ECCs

## 2. Results and Discussion

### 2.1. Solid Phase Extraction

All parameters; mainly the type of SPE cartridges and the pH of sample used to elute the pharmaceuticals under investigation were optimized to find the best extraction efficiencies for the target pharmaceuticals. Three different alternatives of SPE cartridges were tested including two polymeric sorbents (Oasis HLB and ENVI-C18) and a polymeric reversed phase sorbent with anion-exchange groups (Oasis MAX). The extraction efficiency results of the Oasis MAX were better than ENVI-C18 in all cases; however, the recovery results of the Oasis HLB were much better for the selected pharmaceuticals (Figure 1A,B). This hydrophilic–lipophilic-balanced reversed phase sorbent is suitable for the extraction of acidic, basic and neutral compounds and it works well at a pH range of 1–14. However, to find the optimal pH for sample analysis, a pH optimization study was performed to find out the recoveries at different pH values (pH 3 and 7). The extraction procedure described previously was applied to the wastewater samples at pH 3 and 7, spiked with the stock solution of the standard pharmaceuticals at 100 ng/L. The samples were then analyzed using LC-ESI-MS/MS, and the recoveries of analytes were calculated (Table 2). The obtained results showed that the optimum solid phase extraction for all the analytes of interest were found to be at pH 3 using Oasis HLB cartridges (Figure 1C). This may be explained by the fact that the basic analytes under investigation are completely ionized at pH 3, except acetaminophen, and this will avoid the coexistence of ionized and unionized species at pH 7, which might affect their recoveries. The recovery results for all the compounds were higher than 84.7%, as summarized in Table 2.

### 2.2. Optimization of Liquid Chromatography and Mass Spectrometry Conditions

To optimize the chromatographic separation of the target compounds, several chromatographic conditions were tried, including a variation of stationary as well as mobile phases. Different non-polar stationary phases were tried, including Aquity BEH C8, Aquity CSH Phenyl-Hexyl and Aquity BEH C18 column. It was found that the best chromatographic separation for the compounds under investigations was achieved using Aquity BEH C18 column with gradient elution in multiple reactions monitoring (MRM) mode.

Mixtures of organic solvents such as acetonitrile and methanol along with acidified water (with formic acid) were tried. Different concentrations (0.05–0.2%) of formic acid were used to promote protonation of the compounds and generate better peak symmetry. However, none of the obtained chromatograms satisfied the criteria of good separation in term of peaks symmetry and resolution using isocratic elution. The best condition for the separation of compounds of interest was achieved by gradient elution, using methanol (mobile phase A) and 5 mM ammonium formate buffer with % 0.2 formic acid (mobile phase B).

Figure 2 shows the separation of the selected pharmaceuticals using the optimized chromatographic conditions, providing symmetrical peaks and adequate resolution allowing quantitative measurements.

The selected compounds were detected by tandem mass spectrometry MRM. Two precursor ions were selected to produce ion transition for each pharmaceutical using positive electrospray ionization mode (+ESI). MS parameters and data acquisition software system specifications were detailed previously and summarized in Table 7. The MRM LC-MS/MS chromatograms resulting from an analysis of pharmaceuticals of interest by positive ionization mode in wastewater samples are depicted in Figure 3.

### 2.3. Matrix Effect

Matrix effect (ME) was evaluated by comparing the response of the target pharmaceuticals under investigation in the influent and effluent wastewater with respect to their response in pure LC-MS water using the formula [1 − (signal response of spiked matrix/signal response of spiked LC-MS water) × 100. The results shown in Table 3 indicated negligible matrix effect (± ≤ 10).

### 2.4. Linearity and Recovery

The linearity of the proposed method was investigated using LC-MS grade water spiked with five different concentrations of the target compounds by plotting the peak areas of the target pharmaceuticals against their relative concentrations using the linear least square regression method. Attained correlation coefficients were all greater than 0.992.

Recovery (RE) was calculated by comparing the signal intensities of a pre and post SPE spiked samples. Table 4 summarizes the linearity ranges, LOQs, LODs, and recovery percentages. The generated calibration curves suggest that the linearity, LOQs, LODs and recoveries of the proposed LC-MS/MS method satisfy the acceptance criteria of analytical methods for all pharmaceuticals under investigation [52].

### 2.5. Precision

The precision of the method was determined by the repeated intra-day (n = 5) and inter-day (n = 15) analysis of spiked LC-MS grade water at concentrations levels of 15 ng/L and 750 ng/L. The precision of the method was expressed as the relative standard deviation (RSD%) of replicate measurements; the calculated intra- and inter-day precision data are summarized in Table 5.

To ensure that the analytes are stable during the analysis period, a volume of LC-MS grade water was spiked to achieve analyte concentrations of 15 ng/L and 750 ng/L, stored overnight at 4 °C in a glass bottle, and analyzed after 7 days. Furthermore, the stability of the sample in the extracted solvent was tested for additional 5 days in an auto sampler at 4 °C, to ensure that the analytes remain stable during the analysis.

### 2.6. Determination of Target Pharmaceuticals in Wastewater Samples

Composite influent and effluent wastewater samples, collected from Sharjah STP, were analyzed to assess the occurrence of pharmaceuticals and their concentrations. Each sample was analyzed five times utilizing the optimized SPE-LC-MS/MS method. A total of ten pharmaceuticals of different drug classes were detected and quantified at significant concentrations. Concentrations of the selected pharmaceuticals were calculated using the linear regression equations of the relative calibration curves. Table 6 summarizes the concentrations of target analytes in influent and effluent wastewater samples as well as the achieved percent removals of pharmaceuticals in the STP under study.

Generally, removal of organic pollutants at STPs is a complex process with many plausible mechanisms. Pharmaceuticals concentration in the environment are governed by biotic and abiotic (sorption, photo-degradation and hydrolysis) factors and their removal patterns and mechanisms in STPs may be affected either by the specific treatment process employed or by their individual physicochemical properties such as water solubility, volatilization tendency, adsorption kinetics, and degradation potential [53]. In the conventional activated sludge process, removal of pharmaceuticals and personal care products (PPCP) is mainly attributed to two mechanisms, namely sorption onto the particulate phase and biodegradation. During wastewater treatment, PPCPs and their metabolites can partition between the solid/particulate phase and the aqueous phase, depending on their hydrophobicity. Generally, hydrophilic and water-soluble PPCPs show less tendency to sorb onto the solid/particulate phase and are not likely to be detected in sludge. On the other hand, PPCPs with low hydrophobicity, expressed as the octanol–water partition coefficient (K_ow_), are likely to be present in treated effluents of STP if they are resistant to microbial degradation. Several studies have exhibited that in general, compounds with log K_ow_ < 3.0 are not expected to be adsorbed significantly to the particles, thus displaying low removal efficiencies in STPs [54]. On the other hand, compounds with relatively higher log K_ow_ values are expected to exhibit higher removal efficiencies [53,55,56].

For the STP under study, pharmaceutical removals ranged from 31–96%. Acetaminophen level was measured at the highest concentration of 145 ng/L, 250 ng/L and 5,235 ng/L in the influent and effluent samples, respectively, which may be attributed to the frequent prescription of acetaminophen for its antipyretic and analgesic activity. However, a high elimination percentage (96%) of this pharmaceutical was achieved in the treatment system similar to previously published studies [57,58,59,60]. Although acetaminophen exhibits a low log K_ow_ value (0.46), it was shown to be quickly photodegraded in STP effluents [47].

A total of seven antibiotics, including four sulfa-drugs, two fluoroquinolones and one macrolide, were detected in the influent and effluent samples. Among the antibiotics, fluoroquinolone (ciprofloxacin) exhibited the highest concentration with a treatment efficiency of 37%. For sulfa-drugs, the concentration of sulfadiazine was the highest with a partial elimination rate of 40%. The antipsychotic drug risperidone was detected at concentration of 13 ng/L in the effluent samples, with a significant elimination rate of 95%, potentially attributed to its log K_ow_ value of 3.27. Finally, the macrolide antibiotic, erythromycin, and the β-blocker metoprolol were detected at concentrations equivalent to 541 ng/L and 62 ng/L in the effluent, respectively, showing the lowest elimination rates (≤32%) of the treatment system.

## 3. Materials and Methods

### 3.1. Chemicals and Reagents

High purity (>98%) sulfapyridine, sulfadiazine, sulfamethoxazole, metoprolol, sulfamethazine, ciprofloxacin, ofloxacin, risperidone, erythromycin, and acetaminophen were purchased from Sigma-Aldrich (Darmstadt, Germany). Methanol (LC-MS Grade), acetone (HPLC-Grade), water (LC-MS grade water), formic acid (LC-MS Grade), ammonium formate 99%, and hydrochloric acid 37%, were purchased from Sigma-Aldrich (Darmstadt, Germany). Wastewater samples were collected from the influent and effluent of Sharjah central STP.

### 3.2. Sampling Site and Sample Collection

Composite samples from the influent and effluent wastewater of Sharjah STP were collected into dry amber glass bottles pre-rinsed with methanol and LC-MS grade water. Wastewater samples were collected every 90 min, then composited to accommodate for variations in wastewater flows and ECC concentrations at varying sampling episodes; thus, providing samples with better representation. Sharjah STP employs primary (sedimentation), secondary (activated sludge) and sand filters as tertiary treatment, in addition to chlorination for disinfection. All collected samples were properly sealed and transferred to the lab by icebox. In the laboratory, the samples were filtered under vacuum through 0.7 µm glass fiber filters and kept at 4 °C in the dark for a maximum period of 1 week until extraction.

### 3.3. Solid Phase Extraction

Literature surveys suggest that pre-concentration and purification of target pharmaceutical compounds from complex environmental matrices are usually achieved using off line SPE. Wide range of sorbents are commercially available and may be selected based on the degree of polarity of the analytes. The following solid phase extraction cartridges: Oasis HLB (6 mL, 200 mg) and Oasis MAX (6 mL, 150 mg) were obtained from Waters (Milford, MA, USA), and Supelclean ENVI-C18 (6 mL, 500 mg) was purchased from Sigma-Aldrich (Darmstadt, Germany). Glass microfiber filters (0.7 µm pore size, 47 mm diameter) were obtained from LLG Labware (Meckenheim, Germany) and nylon membrane filters (0.45 µm pore size, 47 mm diameter) were from Whatman (Mainstone, UK). The selection of these cartridges was based on their polarity, which suits the physicochemical properties of the compounds under investigation.

In order to optimize an efficient extraction method of the selected compounds, influent and effluent wastewater samples collected from Sharjah STP as well as analyte-free LC-MS grade water were used. The collected wastewater samples were filtered under vacuum using 0.45 µm membrane filters (Whatman, Mainstone, UK) while 0.7 µm membrane filters were used for LC-MS grade water. Filtrates were acidified using 1 M HCl to pH 3 as this pH level exhibited higher recoveries based on a pH optimization experimentation performed on wastewater samples at pH 3 and 7, spiked with 100 ng/L solution of the standard pharmaceuticals.

The performance efficiency of three different cartridges were tested, namely, Oasis HLB (6 mL, 200 mg), a hydrophilic-lipophilic-balanced reversed phase sorbent; Oasis MAX (6 mL, 150 mg), a polymeric reversed phase sorbent with anion-exchange groups; and Supelclean ENVI-C18 (6 mL, 500 mg), a polymeric reversed phase C18 endcapped, and the recovery results were assessed and compared.

The SPE cartridges were pre-conditioned using 3 mL acetone, 3 mL mixed methanol and LC-MS grade (adjusted to pH 3 using 1 M HCl) at flow rate of 3 mL/min. The filtered samples were then passed through the cartridge at 15 mL/min using a vacuum manifold system (Waters) connected to a vacuum pump. The SPE cartridges were then washed twice with a solution (5 mL) of LC-MS grade water: methanol (ratio 95:5) at flow rate of 2 mL/min. and dried under vacuum for 30 min. The elution was then performed with 3 mL methanol three times followed by 3 mL methanol with 0.3% formic acid. The final extracts were mixed and dried using Genevac system (EZ-2 Plus), and then reconstituted in 1 mL of LC-MS grade water: ACN (90:10) prior to LC-MS/MS analysis.

### 3.4. Liquid Chromatography and Mass Spectrometry Conditions

The compounds of interest were analyzed using Waters Acquity^®^ UPLC H-Class-Xevo TQD (Triple Quadrupole Mass Spectrometer) system (Milford, MA, USA) equipped with electrospray ionization (ESI). Chromatographic separation of the target compounds was achieved on Acquity^®^ BEH C18 column (1.7 µm, 2.1 mm × 150 mm) using gradient elution as following: Solvent A was methanol while Solvent B was 0.2% formic acid in 5 mM ammonium formate. The gradient was started with 100% B (*v*/*v*) followed by a 15 min to 50% B; another 0.5 min gradient to 30% B; followed by a 4.5 min gradient to 0% B; during 1 min, steep linear gradient to 100% B; and the column was equilibrated by 100% B for 2 min prior to the next analysis. The flow rate was 0.2 mL/min and the injection volume was 10 μL, while the column was conditioned at 35 °C and the auto sampler performed at 4 °C.

The mass spectrometric analysis was performed in multiple reaction monitoring (MRM) with positive and negative ionization modes. The MS parameters were as follows: Dwell time was 0.02 s and nitrogen was used as a desolvation gas at a flow rate of 600 L/h. The ionization source conditions were as follows: desolvation temperature 350 °C; source temperature 150 °C; collision gas (argon) flow 0.1 mL/min; and capillary voltage 3.0 kV. Compound dependent parameters like parent ion, fragment ion, cone voltage and collision energy were set as shown in Table 7. The parameters of mass analyzer were set as follows: LM1 and HM1 resolution 15 and 15, respectively; ion energy 1; LM2 and HM2 resolution 15 and 15 respectively, and ion energy 2.

The UPLC-MS/MS system control was performed by Lynx software (Waters, Manchester, UK, Version 4.1, SCN 882) and data was processed and analyzed using TargetLynx™ (Waters, Manchester, UK) program.

### 3.5. Method Validation

The optimized SPE LC-MS/MS method was validated for its selectivity, linearity, limit of detection (LOD), limit of quantification (LOQ), intra-and inter day precision, matrix effect, recovery and short time stability following the guidelines of the European Medicines Agency for bioanalytical method validation [61].

The recoveries of the target pharmaceuticals were estimated using LC-MS grade water spiked with two different concentrations equivalent to 15 ng/L and 750 ng/L of the target analytes; the ratios of the concentrations before and after extraction were calculated.

Since wastewater already includes the target compounds, the linearity of the proposed method was evaluated using LC-MS grade water spiked with five different concentration levels. Before injecting the standard solutions, the system was equilibrated for at least 20 min with the mobile phase. Five injections were carried out for each concentration level. The calibration curves were constructed using least square linear regression analysis method. The precision of the method was evaluated as inter and intra-day for five replicates (n = 5) on spiked LC-MS grade water at concentration levels equivalent to 15 ng/L and 750 ng/L of the target analytes. The relative standard deviation values (RSD %) were calculated to express the precision of the proposed method. The limit of detection (LOD) and limit of quantification (LOQ) were calculated based on standard deviation (σ) of the response and slope (s) using the following equations, respectively (LOD = 3.3 σ/s, LOQ = 10 σ/s).

The selectivity of the method was evaluated by replicating the analysis using the multiple reaction-monitoring mode (MRM), which is a highly specific and selective technique, where the compound of interest can be selectively identified and quantified in the presence of other components based on the precursor–ions pair detection.

### 3.6. Analysis of Wastewater

In order to investigate the presence of pharmaceuticals in urban wastewater received and treated at Sharjah STP, influent and effluent wastewater samples from the treatment plant under study were analyzed using the proposed SPE LC-MS/MS method.

## 4. Conclusions

In this study, a highly sensitive and selective SPE UPLC-MS/MS analytical method was developed and validated for the simultaneous determination of the commonly prescribed pharmaceuticals in urban wastewater. The optimized method allowed the simultaneous extraction of 10 pharmaceuticals with different physicochemical properties in one single step using hydrophilic-lipophilic-balanced reversed phase sorbent (Oasis HLB), and the subsequent separation on an Aquity BEH C18 column using gradient elution in multiple reactions monitoring (MRM) mode. The developed method was successfully applied to investigate the occurrence and quantify the amounts of antibiotics and other pharmaceuticals in various wastewater streams of Sharjah STP with a LOD varying between 0.1–4.5 ng/L and LOQ ranging between 0.3–12 ng/L. Moreover, negligible matrix effect was observed in the detection of pharmaceuticals.

The development of such a method would be highly helpful in the assessment and continuous monitoring of such emerging contaminants of concern in the City of Sharjah’s wastewaters with a high degree of accuracy. Findings will be of significance towards supporting decisions to optimize wastewater treatment and reuse strategies, as well as safeguard public and ecological health.

## Figures and Tables

**Figure 1 molecules-24-00633-f001:**
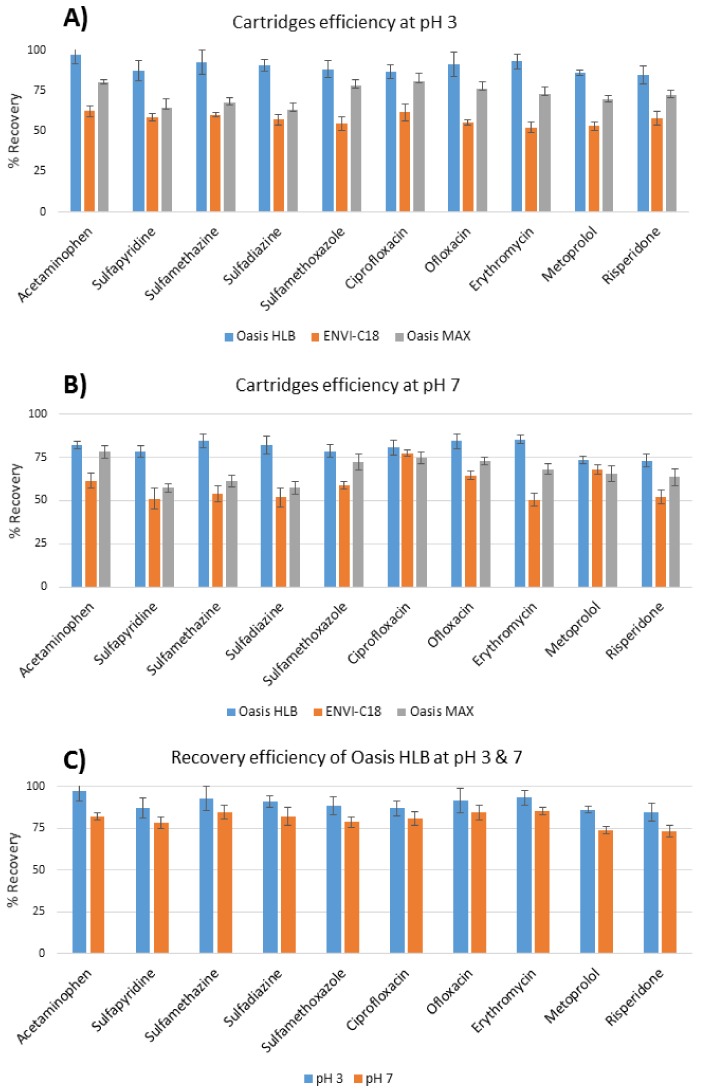
Graph bars showing the effect of pH on the % recoveries of the selected pharmaceuticals at (**A**) pH 3 using Oasis HLB, ENVI-C18 and Oasis MAX cartridges. (**B**) pH 7 using Oasis HLB, ENVI-C18 and Oasis MAX cartridges (**C**) comparison of the recoveries at pH 3 and 7 using HLB oasis cartridge using Oasis HLB, ENVI-C18 and Oasis MAX at two pH values (n = 3).

**Figure 2 molecules-24-00633-f002:**
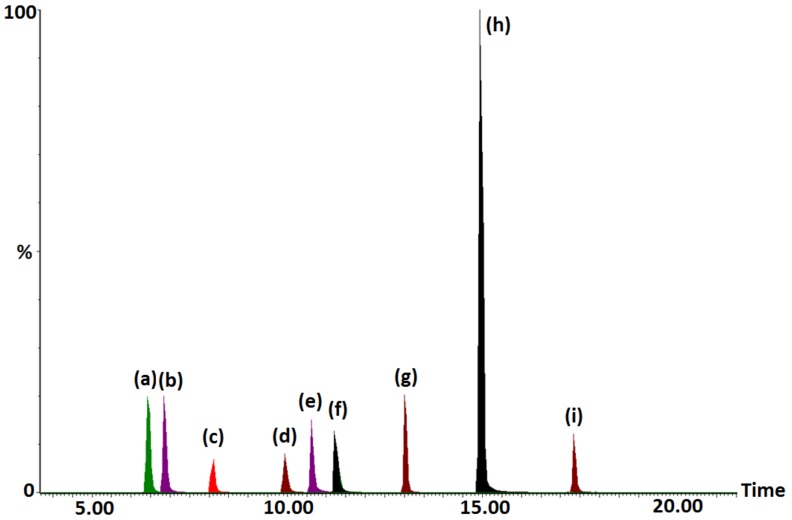
Chromatogram showing the separation of the selected pharmaceuticals using the optimized chromatographic conditions. (**a**) Acetaminophen, (**b**) Sulfadiazine, (**c**) Sulfapyridine, (**d**) Sulfamethazine, (**e**) Ofloxacin, (**f**) (Sulfamethoxazole and ciprofloxacin), (**g**) Metoprolol, (**h**) Resperidone, and (**i**) Erythromycin.

**Figure 3 molecules-24-00633-f003:**
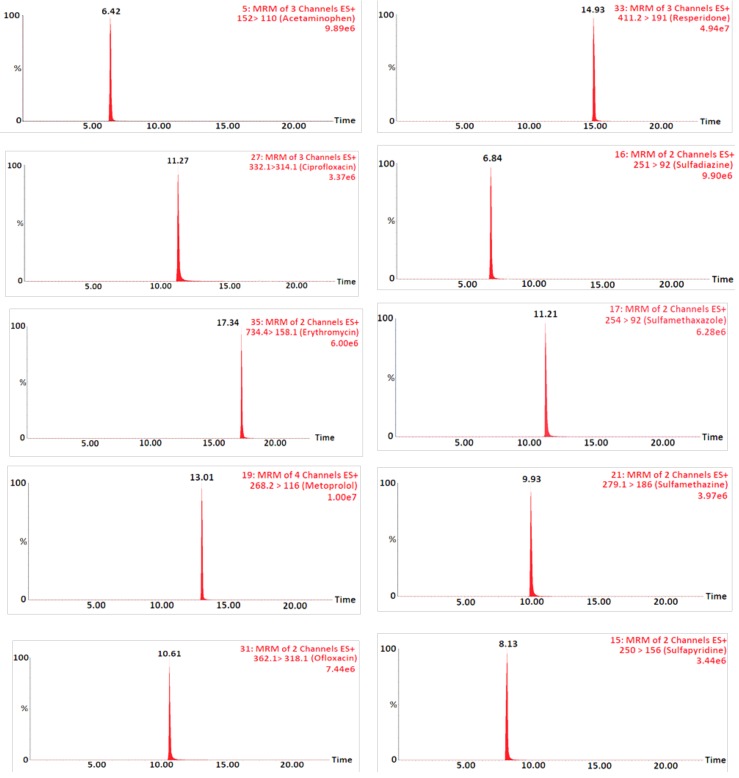
MRM LC-MS/MS chromatograms of the target compounds analyzed by positive ionization mode.

**Table 1 molecules-24-00633-t001:** Physicochemical properties and chemical structures of the pharmaceuticals under investigation.

Compound	Therapeutic Class	Chemical Structure	pK_a_	log P	Reference
Acetaminophen	Analgesic/antipyretic	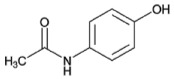	9.38	0.46	[46]
Sulfapyridine	Antibacterial agent	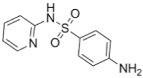	8.43	0.35	[47]
Sulfadiazine	Antibacterial agent	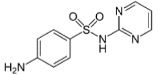	6.36	−0.09	[47]
Sulfamethoxazole	Antibacterial agent	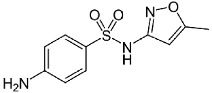	6.16	0.89	[47]
Metoprolol	Beta blocker	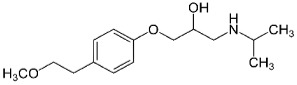	9.4	1.88	[47]
Sulfamethazine	Antibacterial agent	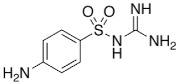	7.59	0.89	[48]
Ciprofloxacin	Antibacterial agent	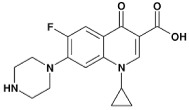	6.09	0.28	[49]
Ofloxacin	Antibacterial agent	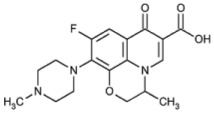	5.97	−0.39	[47]
Risperidone	Antipsychotic	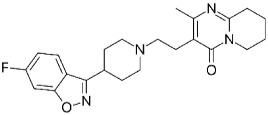	8.76	3.27	[50]
Erythromycin	Antibacterial agent	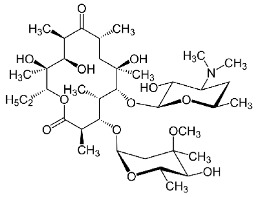	8.88	3.06	[51]

**Table 2 molecules-24-00633-t002:** Recoveries of the selected pharmaceuticals using Oasis HLB, ENVI-C18 and Oasis MAX at two pH values (concentration 15 ng/L, n = 3).

Compound	% recovery at pH 3 (RSD %)			% recovery at pH 7 (RSD %)		
	Oasis HLB	ENVI-C18	Oasis MAX	Oasis HLB	ENVI-C18	Oasis MAX
Acetaminophen	97.2 (5.7)	62.4 (3.4)	80.4 (1.3)	82.1 (2.2)	61.4 (4.4)	78.1 (3.9)
Sulfapyridine	87.3 (6.2)	58.5 (2.3)	64.8 (5.6)	78.3 (3.2)	51.1 (6.1)	57.3 (2.7)
Sulfamethazine	92.7 (7.3)	60.1 (1.2)	67.7 (3.2)	84.6 (4.1)	53.9 (4.5)	61.4 (3.3)
Sulfadiazine	90.8 (3.5)	57.2 (3.3)	63.4 (4.2)	82.2 (5.3)	51.8 (5.3)	57.3 (3.6)
Sulfamethoxazole	88.4 (5.4)	54.6 (4.3)	78.1 (4.1)	78.7 (3.4)	58.7 (2.1)	72.1 (4.5)
Ciprofloxacin	86.9 (4.2)	61.7 (5.1)	81.2 (4.5)	80.8 (4.3)	77.4 (1.8)	74.7 (3.6)
Ofloxacin	91.5 (7.4)	55.5 (1.6)	76.4 (4.3)	84.4 (4.2)	64.5 (2.6)	72.9 (2.1)
Erythromycin	93.3 (4.6)	52.3 (3.2)	73.2 (4.2)	85.3 (2.4)	50.3 (3.7)	68.3 (3.3)
Metoprolol	86.1 (1.8)	53.2 (2.4)	69.8 (2.4)	73.6 (2.2)	68.2 (2.8)	65.4 (4.7)
Risperidone	84.7 (5.6)	58.3 (4.3)	72.2 (3.2)	73.1 (3.5)	52.1 (3.8)	63.5 (4.8)

**Table 3 molecules-24-00633-t003:** Percentage recoveries (n = 3) and matrix effect of spiked LC-MS water, influent and effluent wastewater samples at two different concentration levels (15 and 750 ng/L).

Pharmaceuticals	Spiked conc. ng/L	LC/MS Water %recovery ± RSD	Influent % recovery ± RSD	ME%	Effluent % recovery ± RSD	ME%
Acetaminophen	15	97.2 ± 5.7	98.0±4.5	5	95.0±5.5	3
	750	94.4± 2.4	97.3±3.9		92.3±4.7	
Sulfapyridine	15	87.3 ± 6.2	85.4±6.5	7	95.4±7.5	1
	750	92.1± 3.2	89.4±4.5		91.4±3.6	
Sulfamethazine	15	92.7 ± 7.3	86.3 ± 6.1	8	97.7 ± 6.4	−3
	750	89.9± 4.5	93.6± 3.5		104.3± 9.1	
Sulfadiazine	15	90.8 ± 3.5	97.6 ± 6.1	4	96.1± 9.3	5
	750	87.5± 3.1	79.2± 6.2		84.3± 6.6	
Sulfamethoxazole	15	88.4 ± 5.4	97.6 ± 6.9	−1	90.1 ± 6.4	7
	750	85.2± 3.5	93.6± 3.5		89.6± 3.6	
Ciprofloxacin	15	86.9 ± 4.2	99.1 ± 7.2	3	81.6 ± 9.4	2
	750	88.6±2.8	85.3±4.9		90.8±5.9	
Ofloxacin	15	91.5 ± 7.4	100.5 ± 5.3	6	87.4 ± 8.2	−3
	750	86.3±4.7	93.6±9.4		92.9±6.3	
Erythromycin	15	93.3 ± 4.6	101.9 ± 6.2	9	84.3 ± 6.9	9
	750	91.1±2.7	88.3±3.8		96.3±6.3	
Metoprolol	15	86.1 ± 1.8	80.9 ± 3.3	8	91.6 ± 6.2	8
	750	88.4±3.3	95.3±5.6		80.4±8.1	
Risperidone	15	84.7 ± 5.6	91.3 ± 3.9	10	79.5 ± 6.1	3
	750	86.2±2.6	94.2±4.2		82.8±4.1	

**Table 4 molecules-24-00633-t004:** Linearity ranges, LOQs, LODs and % recoveries of the selected compounds.

Analyte	Linearity Range (ng/L)	Correlation Coefficient (r^2^)	LOD (ng/L)	LOQ (ng/L)	Recovery ± RSD% (n = 5)	
					15 ng/L	750 ng/L
Acetaminophen	5-2500	0.9976	0.1	0.3	97.2 ± 5.7	94.4 ± 2.4
Sulfapyridine	5-1000	0.9968	0.4	1.2	87.3 ± 6.2	92.1 ± 3.2
Sulfamethazine	5-1000	0.9975	0.9	2.9	92.7 ± 7.3	89.9 ± 4.5
Sulfadiazine	5-1000	0.9943	1.3	3.7	90.8 ± 3.5	87.5 ± 3.1
Sulfamethoxazole	5-1000	0.9923	1.4	2.1	88.4 ± 5.4	85.2 ± 3.5
Ciprofloxacin	5-1000	0.9993	1.5	4	86.9 ± 4.2	88.6 ± 2.8
Ofloxacin	5-1000	0.9996	1.1	3.6	91.5 ± 7.4	86.3 ± 4.7
Erythromycin	5-1000	0.9989	1.5	5.0	93.3 ± 4.6	91.1 ± 2.7
Metoprolol	5-1000	0.9969	1.0	3.2	86.1 ± 1.8	88.4 ± 3.3
Risperidone	5-2500	0.9984	1.2	3.6	84.7 ± 5.6	86.2 ± 2.6

**Table 5 molecules-24-00633-t005:** The intra- and inter-day precision of the optimized method.

Analyte	Intra-day RSD % (n = 5)		Intra-day RSD % (n = 15)	
	15 ng/L	750 ng/L	15 ng/L	750 ng/L
Acetaminophen	4.2	4.4	6.1	6.7
Sulfapyridine	3.2	4.3	4.7	5.3
Sulfamethazine	4.3	2.6	8.6	6.4
Sulfadiazine	2.2	4.2	6.8	4.7
Sulfamethoxazole	3.8	3.5	7.5	4.2
Ciprofloxacin	5.1	3.7	7.7	5.4
Ofloxacin	3.6	2.2	6.2	2.6
Erythromycin	2.4	4.2	3.2	4.9
Metoprolol	3.9	2.4	5.3	3.2
Risperidone	4.5	3.9	5.5	4.6

**Table 6 molecules-24-00633-t006:** Concentrations of target contaminants in influent and effluent wastewaters of Sharjah STP and their removal efficiencies.

Analyte	Mean Concentration (ng/L) (n = 5)		
	Influent	Effluent	Removal
Acetaminophen	145250	5235	96
Sulfapyridine	252	99.9	60
Sulfamethazine	24.0	11.0	53
Sulfadiazine	720	433	40
Sulfamethoxazole	161	75.0	54
Ciprofloxacin	863	543	37
Ofloxacin	846	511	40
Erythromycin	785	541	31
Metoprolol	92	62.0	32
Risperidone	245	13.0	95

**Table 7 molecules-24-00633-t007:** MRM and MS parameters for all analyzed compounds.

Pharmaceuticals	Classification	Precursor Ion (*m*/*z*)	Products Ions (*m*/*z*)	Retention Time (min)	CE (V)	CV (V)
Acetaminophen	Analgesic	152.0	110.0	6.47	22	42
			65.0		26	
Sulfapyridine	Antibacterial	250.0	156.0	8.13	16	27
			108.0		25	
Sulfadiazine	Antibacterial	251.0	92.0	6.90	27	23
			156.0		15	
Sulfamethoxazole	Antibacterial	254.0	92.0	11.31	26	27
			156.0		16	
Metoprolol	β-blockers	268.2	116.0	13.06	18	30
			133.0		24	
Sulfamethazine	Antibacterial	279.1	186.0	9.98	16	30
			92.0		28	
Ciprofloxacin	Antibacterial	332.1	314.1	11.33	22	30
			288.1		18	
Ofloxacin	Antibacterial	362.1	318.1	10.66	26	30
			261.1		20	
Risperidone	Anti-depressant	411.2	191.0	14.98	30	40
			110.0		50	
Erythromycin	Antibiotic	734.4	158.1	17.39	32	25
			576.3		40	

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
