# Peer review of "Simultaneous Determination of Pharmaceuticals by Solid-phase Extraction and Liquid Chromatography-Tandem Mass Spectrometry: A Case Study from Sharjah Sewage Treatment Plant"

_molecules, 2019, doi:10.3390/molecules24030633_

Round 1
Reviewer 1 Report
The manuscript entitled " Simultaneous determination of pharmaceuticals by solid-phase extraction and liquid chromatography–tandem mass spectrometry: a case study from Sharjah sewage treatment plant" reports on the application of commercial cartridges for the analysis of contaminants in water
Some critical issues should be clarified before a possible publication.
Line 42 authors should add some recent papers carried out on contaminants in water
Line 54 authors should add a reference Journal of Separation Science Volume 37, Issue 20, 1, Pages 2882-289154 about personal care products
Lines 126-127-128 The authors used and compared different cartridge with different amount of stationary phase. The authors should be clarified their choice since it is not possible carried out a comparison in this way.
Line 142 The same solvents were used for the three SPE cartridges. Why the authors did not optimezed a specific method and specific solvents for every type of cartyridge?
In table 2, please add significant digits. Data should be presented in a rigorous way.
Line 185 Why the method precision was calculated on MS grade water. Usually it is common and rigorous calculate it on fortified real samples.
Line 188 The same issue: why LOD e LOQ were calculated in solvent and no in the real sample?
Line 215 Please justify the low value of recovery. Maybe the authors had some elution problem? or a small interaction with stationary phase or probably the cartdrige was overloaded? Did The authors employed an elution curve in order to evaluate these issues?
Line 229 Why organic modifier was different in the two mobile phases (A and B)
Line 236 The obtained peaks are not really good. They appear large, tailed and two analytes probably were co-eluted.
Line 255 A table with matrix effect is missing. The authors should be add. If they do not have this table falls on the assumption of method validation. Moreover, the method validation was carried out on MS grade water. This is not rigorous, if the matrix effect is high it is not a right choice.
Line319 Please explain the innovation of this method compared to other already present method in the literature. Where it is possible highlight this. In the present literature anyone analyzed simoultaneously these analytes??
Author Response
Author responses to reviewer 1 comments
Thank you so much for your valuable comments that helped us improve our article.
In the manuscript, the corrected parts have been highlighted in yellow color, and the author point-by-point responses to reviewer comments are highlighted in blue color
Response to Reviewer 1 comments
1- Line 42 authors should add some recent papers carried out on contaminants in water
Response: Three recent articles were added. Page1, line 41. (ref. 1-7)
2- Line 54 authors should add a reference Journal of Separation Science Volume 37, Issue 20, 1, Pages 2882-289154 about personal care products
Response: The mentioned reference has been cited. Line 54.
3- Lines 126-127-128 The authors used and compared different cartridge with different amount of stationary phase. The authors should be clarified their choice since it is not possible carried out a comparison in this way.
Response: this point has been clarified in the manuscript. Please refer to line 143-145.
4- Line 142 The same solvents were used for the three SPE cartridges. Why the authors did not optimezed a specific method and specific solvents for every type of cartridges?
Response: Since the question in matter was the comparison of cartridges efficiency and their extraction power, one type of solvent (water) therefore was used to evaluate the extraction efficiency of the cartridges. However, since the ionization state of the compounds under investigation is pH and pKa dependent, the extraction was tested at two different pH values under aqueous condition where it was easier to adjust and control the pH at certain values, and the results showed superior recovery values at pH 3.
5- In table 2, please add significant digits. Data should be presented in a rigorous way.
Response: The numbers in table 2 were modified accordingly. (page 6)
6- Line 185 Why the method precision was calculated on MS grade water. Usually it is common and rigorous calculate it on fortified real samples.
Response: The precision of the method was evaluated using LC-MS grade water to exclude the coexistence of pharmaceuticals as possible interferences. The fortified solvent can be used; however, other methods reported the use of pure solvents including methanol, acetonitrile and LC-MS grade water. The following references is an example of that: 1) Talanta 74 (2008) 1463–1475. “Development and optimization of a single extraction procedure for the LC/MS/MS analysis of two pharmaceutical classes residues in sewage treatment plant”.
2) J Chromatogr A. 2006, 1114(2):224-33, Determination of pharmaceuticals of various therapeutic classes by solid-phase extraction and liquid chromatography-tandem mass spectrometry analysis in hospital effluent wastewaters.
7- Line 188 The same issue: why LOD e LOQ were calculated in solvent and no in the real sample?
Response: the same approach, as mentioned in the previous point, was applied in the determination of LOD and LOQ.
8- Line 215 Please justify the low value of recovery. Maybe the authors had some elution problem? or a small interaction with stationary phase or probably the cartridge was overloaded? Did The authors employed an elution curve in order to evaluate these issues?
Response: Since the purpose of this study was to compare the extraction efficiency of different cartridges and based on the recovery results, it was found that Oasis HLB provided the best % recovery (84.7-97.2 %), which indicates high recovery values.
9- Line 229 Why organic modifier was different in the two mobile phases (A and B)
Response: The mobile phase that was used in this study includes only one organic modifier (methanol –mobile phase A) in a gradient elution mode, while the aqueous mobile phase (B) includes formic acid to facilitate the protonation of the compounds under investigation and that’s tremendously improved the method sensitivity and peak symmetry.
10- Line 236 The obtained peaks are not really good. They appear large, tailed and two analytes probably were co-eluted.
Response: The peak symmetry was evaluated by calculating the tailing factor and number of theoretical plates and the results were found to be within the acceptance criteria of system suitability parameters. We agree with reviewer’s 1 comment concerning the co-elution of two compounds only in peak F (Sulfamethoxazole and ciprofloxacin), in figure 2, however, the mass spectrometric analysis were performed in multiple reactions monitoring (MRM) selecting two precursor ions to produce ion transition for each pharmaceutical using positive electrospray ionization (+ESI) mode. The advantage of using this mode is the discrimination between two peaks if they co-elute together.
11- Line 255 A table with matrix effect is missing. The authors should be add. If they do not have this table falls on the assumption of method validation. Moreover, the method validation was carried out on MS grade water. This is not rigorous, if the matrix effect is high it is not a right choice.
Response: A table showing the matrix effect of spiked LC-MS water, influent and effluent wastewater samples at two different concentration levels ( 15 and 750 ng/L) was inserted in page 11 (table 3) and an explanation of the matrix effect measurement was highlighted in a separate paragraph (page10, line 287-292).
12- Line319 Please explain the innovation of this method compared to other already present method in the literature. Where it is possible highlight this. In the present literature anyone analyzed simoultaneously these analytes??
Response: A paragraph explaining the novelty of this study was added in the introduction section (page3, line 97-103)
Reviewer 2 Report
The manuscript is well written and organized and the presented methods are adequately validated. To my opinion the paper is useful for the readers and I support its publication after revision.
Comments
Introduction
1. It will be good for the readers to know the reasons of the selection of these specific drugs (presented in table 1) for the study. Why is it so important to study Acetaminophen (paracetamol), Sulfapyridine, Sulfadiazine, Sulfamethoxazole, Metoprolol, Sulfamethazine, Ciprofloxacin, Ofloxacin, Risperidone, and Erythromycin as emerging contaminants of concern? Please take under consideration their pharmacological action, drawbacks of their use and separate these drugs into specific classes according to their pharmacology so as to justify this. Please organize the introduction accordingly.
2. I would expect a discussion on the choice of this specific Acquity® BEH C18 analytical column.
3. Line 188: LOD and LOQ were determined using the calibration curve of each target compound. Please explain in more detail how you derived LOQ & LOD from the calibration curves.
4. Line 189: Please explain in more detail how you derived selectivity.
5. Line 211-214. Please provide a bibliography on how you calculated the ionization state of the analytes.’
6. Specify the reasons of the specific concentration ranges presented in table 4 for each one of the analytes.
Author Response
Author responses to reviewer 2 comments
Thank you so much for your valuable comments that helped us improve our article.
In the manuscript, the corrected parts have been highlighted in yellow color, and the author point-by-point responses to reviewer comments are highlighted in blue color
Response to Reviewer 2 comments
1- It will be good for the readers to know the reasons of the selection of these specific drugs (presented in table 1) for the study. Why is it so important to study Acetaminophen (paracetamol), Sulfapyridine, Sulfadiazine, Sulfamethoxazole, Metoprolol, Sulfamethazine, Ciprofloxacin, Ofloxacin, Risperidone, and Erythromycin as emerging contaminants of concern? Please take under consideration their pharmacological action, drawbacks of their use and separate these drugs into specific classes according to their pharmacology so as to justify this. Please organize the introduction accordingly.
Response: This point was addressed by adding a new paragraph in the introduction section highlighted in yellow color (page2, line 70-77).
2- I would expect a discussion on the choice of this specific Acquity® BEH C18 analytical column.
Response: This matter was addressed in the results and discussion section, please refer
to page 8, line 257-261.
3- Line 188: LOD and LOQ were determined using the calibration curve of each target compound. Please explain in more detail how you derived LOQ & LOD from the calibration curves.
Response: This point was addressed. Please refer to page 6, line 212-214.
4- Line 189: Please explain in more detail how you derived selectivity.
Response: This point was addressed. Please refer to line page 6, 216-219.
5- Line 211-214. Please provide a bibliography on how you calculated the ionization state of the analytes.’
Response: The ionization of the target compounds depends on their pka’s and the pH of the solvent. Please refer to table 1, where the pka’s of the target compounds are listed with their references.
6- Specify the reasons of the specific concentration ranges presented in table 4 for each one of the analytes.
Response: the concentration range of each analyte was established based on the obtained correlation coefficient (r2) of the proposed concentrations as an indicator for an acceptable linearity.